# Effects of Changes in Environmental Color Chroma on Heart Rate Variability and Stress by Gender

**DOI:** 10.3390/ijerph19095711

**Published:** 2022-05-07

**Authors:** Jiyoung Oh, Heykyung Park

**Affiliations:** 1Research Institute of Ecology, Pusan National University, Busan 46241, Korea; ojy1245@pusan.ac.kr; 2Department of Interior Architecture, Inje University, Gimhae-si 50834, Korea

**Keywords:** environmental color, chroma, heart rate variability, stress emotion

## Abstract

With increasing time spent indoors during the coronavirus disease pandemic, occupants are increasingly affected by indoor space environmental factors. Environmental color stimulates human vision and affects stress levels. This study investigated how changing environmental color chroma affected heart rate variability (HRV) and stress. The HRV of nine males and fifteen females was measured during exposure to 12 color stimuli with changes in chroma under green/blue hues and high/low-value conditions, and a stress assessment was performed. The effect of chroma on the HRV of males and females was verified, but the interaction effect between chroma and gender was not. ln(LF) and RMSSD were valid parameters. ln(LF) of males and females decreased as chroma increased under the green hue and low-value conditions; RMSSD was reduced as chroma increased in the blue hue and low-value conditions. ln(LF) decreased as chroma increased under blue hue and high-value conditions in males. Color–stress evaluation revealed that the higher chroma under high-value conditions, the more positive the stress emotion, and the lower the chroma under low-value conditions, the more negative the stress emotion. As chroma increased under low-value conditions, color is a stress factor; for men, this effect was more evident in the blue hue.

## 1. Introduction

Urbanization, industrialization, and more competitive social atmospheres caused by rapid social structural changes are creating unhappy societies and reducing self-esteem [1]. This social atmosphere increases exposure to stress, which has a negative effect on psychological and physiological human domains [2]. The ongoing coronavirus disease (COVID-19) pandemic has made clear how much stress diseases and changes in social structures can cause among individuals and countries. National measures to curb the pandemic have caused physical and mental stress problems [3,4,5,6]. Particularly, self-isolation, a strategy to minimize exposure to COVID-19, has increased depression and resulted in productivity loss due to people changing their living spaces into studying and working spaces [7].

Korea has excellent public health facilities, with more hospital beds and medical equipment than the Organisation for Economic Co-operation and Development (OECD) average. It also has the highest number of outpatient department visits per capita among OECD countries [8]. Accordingly, Korea has a lower mortality rate from major diseases than other OECD countries and is a major medical tourism destination in Asia [9]. Korea is also recognized as an exemplary case in slowing the pace of the pandemic without enforcing regional lockdowns due to the active response and strict personal hygiene rules introduced by the government following the outbreak of COVID-19 [10]. Nevertheless, South Koreans are under more stress than people in Japan, China, the United States, the United Kingdom, and Germany [11]. Additionally, those working in the medical sector have particularly high stress levels [12,13]. The pandemic acted as an environmental factor that caused stress. Studies on the relationship between the pandemic and stress reported that during the pandemic, anxiety, depression, and stress have been higher in young people than in the elderly and higher in females than in males; furthermore, it has caused negative psychological effects [14,15,16,17]. This report suggests that the stress stimulus caused by the external environment may appear differently depending on age and gender. On the physiological stress response according to gender, Dishman et al. (2000) determined that there is no relationship between heart rate variability (HRV), which is used as a measuring indicator of stress, and gender [18]; Madden & Savard (1995) verified this lack of relationship [19]. However, Buchanan et al. (2010) found that cortisol responses to psychological stress differed by gender [20], and Traustadóttir et al. (2003) revealed a gender difference in psychological stress due to higher cortisol in older males than in older females [21]. In other words, there is an observable difference between males and females when responding to physiological and psychological stress. As the time spent in specific spaces increased during the pandemic, the factors of the spatial environment affected the occupants’ stress [22], and environmental color, as an environmental factor with strong visual stimulation, affected the physiology and psychology of occupants [23]. Jonauskaite et al. (2020) clinically verified that occupants perceive colors through sight while resting and that the effects of these colors can impact stress and anxiety levels [24]. Therefore, environmental color is an environmental factor in the stress level of space users, and the stress, anxiety, and depression levels of space users vary depending on the colors in the surrounding spatial environment [24,25,26,27]. If environmental color is an environmental factor that affects the occupant’s physiological and psychological aspects, the differences between physiological and psychological stress in males and females need to be understood. However, previous studies on the relationship between environmental colors and the stress of occupants commonly mention the limitation of insufficient evidence-based research [28,29]; therefore, conducting empirical research on the effect of environmental colors on the physical and psychological stress level of occupants through follow-up studies is necessary.

Although many studies have researched the effect of different colors on the autonomic nervous system, fewer studies have focused on the influence of changes in value (the lightness of the color) or chroma (the intensity of the color). Nonetheless, scholars have consistently mentioned that differences in chroma and hue affect the physiological and psychological states of occupants. In a study by Kwallek et al. (1996) [30], female workers indicated more depression, confusion, and anger in low-chroma offices, whereas male workers reported more depression, confusion, and anger in high-chroma offices. This demonstrated that the chroma of environmental colors is a major variable that influences the psychological state of occupants. Öztürk et al. (2012) [31] confirmed that work accuracy was higher in chromatic schemes than in achromatic schemes, and occupants found chromatic schemes to be more attractive and satisfactory. The chroma of environmental color affects the efficiency and evaluation of workers. Furthermore, Suk et al. (2009) [32] reported that hue, value, and chroma affect emotional responses, with chroma having a stronger effect. Zielinski (2016) [33] validated that changes in chroma affect skin conduction responses and argued that high-chroma colors are better for capturing attention and are a significant factor in color perception. Although these previous studies verified that chroma is a significant factor that influences the physiology and psychological states of occupants, research into how the chroma conditions of environmental colors affect the stress level of occupants is still in its preliminary stages. Therefore, because people are spending more time in spatial environments during the ongoing pandemic, studies should investigate environmental color chroma conditions that have a positive effect on reducing the stress level of occupants. Most previous studies performed experiments using colors close to the primary colors as stimuli, which differ from actual environmental color conditions, so they have limitations in terms of applying the results to a real-world environment. Similar to the study by Oh et al. [23], the present study used colors corresponding to the actual environment instead of primary colors as experimental stimuli to derive empirical research results applicable to the environment.

The objective of this study was to evaluate the stress levels of males and females in accordance with the various color environments by utilizing HRV analysis. HRV was chosen to measure the stress of occupants; HRV objectively evaluates stress by measuring the activity of the sympathetic and parasympathetic nerves constituting the autonomic nervous system, making it a suitable index to measure physiological stress responses. HRV measurement is a non-invasive method that does not entail any penetration of the human body and has advantages such as placing a lighter psychological burden on the subject and having a short measurement time. A stress assessment using adjectives was conducted and employed as a psychological stress response indicator to supplement HRV [34,35,36,37]. Through this, the current study aimed to provide evidence-based research findings for environmental color configurations that can decrease the stress levels of occupants.

## 2. Methods

### 2.1. Configuration of Experimental Colors

Although Korea provides high-quality health care through continuous quality improvement in medical services led by the government, the stress level of medical workers is high. This study configured the experimental colors using environmental color data of public health facilities, which are spaces mainly used by Korean health professionals with high stress levels, with the aim of measuring the change of the autonomic nervous system in response to a color stimulus close to the actual color of the environment, rather than primary colors. Although strong color stimulation of primary colors strongly affects the autonomic nervous system, it is difficult to apply any obtained results to real environments. Thus, results derived from color stimuli constructed by measuring color data in the real environment (public health facilities) are more applicable to real environments. We visited 22 public health facilities in Korea to collect color data by measuring the environmental colors with a spectrophotometer (Minolta CM-2500d, Konica Minolta, Osaka, Japan; Table 1) and derived the average values of the color data and used them as a basis to configure the experimental colors.

Environmental color data were collected from 22 public health facilities, including eight general hospitals with more than 1000 beds each (Gachon University Gil Hospital, Seoul National University Bundang Hospital, Seoul National University Hospital, Samsung Medical Center, Seoul St. Mary’s Hospital, Asan Medical Center, Severance Hospital, and Ajou University Hospital), eight elderly care facilities (Gangbuk Silver Welfare Center, Yangcheon Municipal Elderly Care Center, Bohyeon-haengwon Elderly Care Center, Municipal Elderly Care Center, Sinmangae Elderly Care Center, Aegwang Elderly Care Center, Walker Hill Silver Town, and Hyoneung-won Elderly Care Center), and six public health centers (Nowon-gu Public Health Center, Seongbuk-gu Public Health Center, Bupyeong-gu Chungchun Public Health Center, Buk-gu Gangbuk Public Health Center, East City Public Health Center, and Yeonje-gu Jaeban Public Health Center). In total, 554 environmental color data points were collected from these facilities. Table 2 summarizes the results of analyzing the average values of the color data according to the weights of the dominant, complementary, and accent colors applied to the walls (the spatial range where the visual perception of the space users mainly occurs).

The most used Munsell color system expresses color properties by dividing them into hue, value, and chroma when representing color data. Hue indicates the properties of color, and the basic five colors (red, yellow, green, blue, and purple) are expressed as 5R, 5Y, 5G, 5B, and 5P, respectively. The intermediate colors of these colors are expressed as 5RY, 5YG, 5BG, 5PB, and 5RP. Value means the brightness of the color and varies from 0 to 10, in which black, with the darkest brightness, is displayed as 0, and the brightest white is expressed as 10. Chroma represents the intensity of the color; an achromatic color without color is expressed as 0, and the value increases according to the degree of saturation. The maximum value of chroma depends on the hue and value. The color representation of the Munsell color system is expressed in the form of “hue value/chroma” [38]. The experimental stimuli in this study consisted of green and blue hues. The values were based on the average value of the dominant color (8.6) and the average value of the accent color (6.3). Chroma was increased by three degrees on the basis of the average chroma (1.8) of the dominant color to configure low chroma (1.8), medium chroma (4.8), and high chroma (7.8) [23]. Figure 1 shows the 12 color samples configured according to these criteria and utilized to measure the HRV and assess the stress levels of males and females according to changes in chroma.

### 2.2. Research Procedure

Figure 2 summarizes the five steps of this study. In step 1, we explained the outline and the precautions of the experiment through interviews with the subjects while obtaining their written consent to participate in the experiment. In step 2, we performed the Ishihara test for color blindness and color vision. In step 3, we measured HRV under 12 color environments. In step 4, we assessed stress on the color stimuli. In step 5, we conducted a multivariate analysis of variance (MANOVA) analysis on the HRV measurement and stress assessment data to examine how changes in environmental color chroma affected the HRV and emotional evaluation of males and females.

### 2.3. Method of Measuring HRV

For HRV measurement, the participants were randomly exposed to three groups of color stimuli in the order of Group A (four colors corresponding to low chroma), Group B (four colors corresponding to medium chroma), and Group C (four colors corresponding to high chroma). Before starting the experiment, the subjects covered their eyes with an eye patch and rested for 5 min. Participants were exposed to each color stimulus for 2 min 30 s, and HRV was measured for the duration [39,40,41]. Between each stimulus, the participants covered their eyes with an eye patch and rested for 1 min [42]. Behaviors that could affect the autonomic nervous system, such as deep breathing, sneezing, coughing, and yawning, were restricted during the experiment. The HRV measurement monitor was turned on during the HRV measurement period, and only the measurer could see the monitor.

According to the recommendations of the HRV guidelines, a 5-min measurement of HRV per color is methodologically adequate. However, following this guideline would result in an experiment time of over 60 min, which the researchers judged would increase stress in the participants, making it impossible to obtain accurate data. The measurement time was reduced by considering effective parameters in ultra-short-term measurements. Recently, several mobile health-care devices have been used to estimate stress by measuring HRV; thus, there have been many discussions on HRV parameters that are effective for ultra-short-term measurements of less than 5 min. Generally, researchers agree that RMSSD is a valid parameter for ultra-short-term measurements [43]. Salahuddin et al. (2007) [44], Castaldo et al. (2019) [41], Baek et al. (2015) [45], and Marek Malik et al. (1996) [46] verified that the low frequency (LF) and high frequency (HF) have been normalized. Therefore, this study used a log of low-frequency HRV (ln(LF)), a log of high-frequency HRV (ln(HF)), and RMSSD as analysis parameters to measure HRV in 12 color environments, as explained below.

LF and HF parameters are data values derived through frequency domain analysis. LF represents a power value between 0.04–0.15 Hz. It simultaneously reflects the activities of the sympathetic and parasympathetic nervous systems and, in long-term measurement, reflects the activity of the sympathetic nerve more [47,48,49]. A decrease in LF reflects a fatigued state, while an increase in LF is known to have a positive correlation with depression, anger, and stress [50]. HF has a power value between 0.15–0.4 Hz. It reflects the activity of the parasympathetic nervous system. The raw data of LF and HF have a distorted value because the variance is large and does not follow a normal distribution. Natural logarithmic values are taken to compensate for this, which are expressed as ln(LF) and ln(HF). RMSSD is a data value derived through time domain analysis. It is the square root value of the mean square of the RR(NN) interval difference. Additionally, it reflects short-term variability in heart rate and the activity of the parasympathetic nervous system [48,49,50]. Under stressful conditions, parasympathetic nerves are reduced, and the HF and RMSSD parameters reflecting parasympathetic activity are decreased [18].

An HRV measuring instrument (uBio Macpa, Biosense Creative Co. Ltd., Seoul, Korea) was used to measure HRV (Table 3, left). Measurement followed the measurement and analysis method of the HRV guidelines presented by the European Society of Cardiology and the North American Society of Pacing and Electrophysiology. It was applied with the autonomic nervous system data standard presented by the HeartMath Research Center in the United States. The instrument can measure 40 to 200 beats per minute (BPM), with an error rate of 2%. The accuracy and reliability of the instrument and analysis software have been proven in past studies [51,52,53]. An analysis software application (Table 3, right) provided raw data by converting HRV measurement data into Excel data, and the mean pulse, complexity, total power, VLF, LF, HF, LF/HF, BPM, SDNN, RMSSD, and stress index parameter data were derived.

### 2.4. Color–Stress Assessment

Color–stress assessment used adjectives expressing opposite emotions to assess the emotions felt when recognizing color samples on a 5-point Likert scale.

The words employed for the color–stress assessment were extracted from four doctoral dissertations related to color appraisal [54,55,56,57]. Table 4 below presents the seven paired adjectives to evaluate stress emotions. These words were evaluated by three experts with a Ph.D. in the field of environmental color.

After the participants took a 3-min break following the HRV measurement, the color–stress assessment was conducted by presenting 12 color samples in the order of G1–G6 and B1–B6. The participants then evaluated their perceived emotions on a 5-point scale. The color samples were the same colors as the environmental colors used in the HRV measurement, and their size was 10 × 10 cm. The assessment was conducted on a gray-tone background, an achromatic color series that increases color perception with a minimum effect on color interference [58].

MANOVA analysis was performed to verify the main effect of color chroma change and gender and the interaction effect between color chroma change and gender on the seven paired adjectives used in the color–stress assessment.

### 2.5. Participants

A total of 24 Korean people (9 male and 15 female) participated in the experiment with an average age of 21.4 years old. According to reports, people in their 20s are the most vulnerable age group to stress as the proportion of patients with mood disorders, depressive disorders, and anxiety disorders in their 20s is overwhelmingly high [59]. People in their 20s reported having high levels of depression, anxiety, and stress, not only in Korea but also in the United States, Spain, China, and Saudi Arabia [60,61,62,63,64]. Therefore, the participants of this study consisted of young people in their 20s. People in their 20s have excellent visual conditions for perceiving colors. It was also easier to accurately measure HRV due to their active physical vitality.

All the participants voluntarily expressed their intention to participate in the experiment after reviewing the contents of the experiment through online and offline notices. Participants were excluded if they were smokers, claustrophobic, mentally ill, obese, or suffered specific chronic illnesses that could affect their autonomic nervous system. Moreover, they were asked to abstain from drinking and obtain at least 6 h of sleep before the experiment. Caffeine intake was also restricted for 3 h before starting the tests.

### 2.6. HRV Measurement Environment

The experimental environment blocked the inflow of natural light and was composed of artificial lighting (Table 5, Osram FHF32SS EX-D, 32W, four ramps, Osram, Munich, Germany). Illuminance was measured twice, once in the morning and once in the afternoon of the experiment, with an illuminance meter (Table 6, Luton LM-81LX). The average illuminance was 359 lux [65], and the dimensions of the experimental space were 6.0 (w) × 3.0 (d) × 2.4 m (h). Additionally, the walls and ceiling were finished with achromatic white matte paint and the floor with achromatic gray tiles. Figure 3 shows that the color space for the HRV measurement had color stimulus materials large enough (1.0 (w) × 0.8 (d) × 1.2 m (h)) to block the vision of the participants. The experimental environment was designed to be similar to an ordinary living space. The illuminance was set to the level of typical office spaces (250–500 lux) [66], and the ceiling received ceiling lighting without any color stimulus.

### 2.7. Statistical Analysis Method

As discussed above, for HRV measurement data analysis, the mean and standard deviation values were derived through frequency analysis centered on ln(LF), ln(HF), and RMSSD parameters to identify changes in overall parameters. MANOVA was performed to verify the main effects of chroma and gender on HRV parameters and the interaction effect between chroma and gender.

After analyzing the color–stress evaluation results, the mean values of the evaluation values of males and females for seven pairs of adjective vocabulary were expressed as a line graph. Additionally, MANOVA was performed to verify the main effects of chroma and gender and the interaction effect between chroma and gender for the seven pairs of adjective vocabulary. The applications used for statistical analysis were Predictive Analytics SoftWare (PASW) Statistics 18 and Microsoft Excel 2019 version.

## 3. Results and Discussion

### 3.1. Results

#### 3.1.1. HRV Results by Gender According to Changes in Environmental Color Chroma

Table 7 shows the mean and standard deviation of ln(LF), ln(HF), and RMSSD parameters by gender for each experimental color. The basic unit of ln(LF) and ln(HF) is ms^2^, and that of RMSSD is ms.

In green hues, except for G1, the RMSSD was slightly higher in females than in males. Similarly, in blue hues, the RMSSD was higher in females than males.

MANOVA statistical analysis was performed on the HRV data before measurement and the three colors with chroma changes to verify whether ln(LF), ln(HF), and RMSSD parameters derived from the HRV measurement exhibit significant differences according to changes in environmental color chroma and gender and to examine the interaction effects (Table 8). To “verify the homogeneity of variances,” the basic assumption for the performance of MANOVA, the M verification was conducted for BOX. As a result, “Before × G1 × G3 × G5” was *p* = 0.520, “Before × G2 × G4 × G6” was *p* = 0.638, “Before × B1 × B3 × B5” was *p* = 0.745, and “Before × B2 × B4 × B6” was *p* = 0.582, which showed no statistical difference in the covariance matrices, thereby confirming that MANOVA is possible.

As seen in Table 8, in MANOVA (wilks’ λ = 0.932, F(9, 209.452) = 0.783, η^2^ = 0.026), as per changes in chroma under G hue-high-value conditions, there were significant differences in ln(LF) (F = 4.667, *p* < 0.05) according to gender. Males showed the highest ln(LF) in color G3, while ln(LF) of females tended to decrease as chroma increased, but the difference was not significant.

In MANOVA (wilks’ λ = 0.908, F(9, 209.452) = 0.945, η^2^ = 0.032), as per changes in chroma under G hue-low-value conditions (G2 × G4 × G6), there were significant differences in ln(LF) (F = 4.484, *p* < 0.05) according to gender. ln(LF) values of males and females tended to decrease as chroma increased. In the case of males, there was a significant decrease in ln(LF) for G6.

In MANOVA (wilks’ λ = 0.912, F(9, 209.452) = 0.895, η^2^ = 0.030), as per changes in chroma under B hue-high-value-conditions (B1 × B3 × B5), were significant differences in ln(LF) (F = 7.233, *p* < 0.01) according to gender. Males recorded the highest ln(LF) value in B1 and ln(LF) decreased as chroma increased. There was almost no change in females.

In MANOVA (wilks’ λ = 0.953, F(9, 209.452) = 0.469, η^2^ = 0.016), as per changes in chroma under B hue-low-value conditions, there were significant differences in RMSSD (F = 4.326, *p* < 0.05) according to gender. The RMSSD of males and females decreased as chroma increased.

The interaction of chroma change with HRV did not vary according to gender, but the interaction with HRV parameters did. Generally, there was a difference in ln(LF) according to gender. There was also a difference in the RMSSD as chroma changed under blue hue-low-value conditions (B2 × B4 × B6).

#### 3.1.2. Stress Assessment Results According to Changes in Environmental Color Chroma

Table 9 shows the average values of the stress assessment to understand how the stress levels of males and females changed according to changes in environmental color chroma.

Under high-value conditions of green and blue hues (G1 × G3 × G5 and B1 × B3 × B5, respectively), stress emotions were perceived to be positive as chroma increased. On the contrary, under low-value conditions of green and blue hues (G2 × G4 × G6, B2 × B4 × B6), stress emotions were perceived to be negative as chroma decreased. Therefore, changes in chroma affect stress levels, and the effect of color is greater than that of gender.

MANOVA statistical analysis was conducted to verify whether there were significant differences in stress emotions according to chroma change and gender and to identify whether there was an interaction effect. Table 10, Table 11, Table 12 and Table 13 show the results.

As an interval scale evaluated stress assessment, a normal distribution test was conducted and confirmed through skewness and kurtosis values. In Hopeless–hopeful, skewness was 0.064 and kurtosis was −0.933, and in Annoying-enjoyable, skewness was 0.193 and kurtosis −0.205. In Tensioned-relaxed, skewness was −0.089, and kurtosis was −0.0619. In Depressed–cheerful, skewness was 0.163, and kurtosis was −0.422. In Boring-fun, skewness was 0.168 and kurtosis −0.594, and in Overwhelming–free, skewness was −0.305 and kurtosis −0.300. In Dislike–like, skewness was −0.112, and kurtosis was −0.362. When the absolute value of the skewness index is 3.0 or less, and the absolute value of the kurtosis index is 8.0 or less, it can be seen as a normal distribution [67]. Therefore, it is confirmed that a dependent variable composed of an interval scale follows a normal distribution.

Before MANOVA, the homogeneity of variances was verified, and “G1 × G3 × G5” was shown as *p* = 0.288, “G2 × G4 × G6” as *p* = 0.749, “B1 × B3 × B5” as *p* = 0.653, and “B2 × B4 × B6” as *p* = 0.027. Overall, no statistical differences were observed in the covariance matrixes, and thus, MANOVA was performed.

Table 10 shows that in MANOVA (wilks’ λ = 0.650, F(14, 120) = 0.2.063, η^2^ = 0.194), as per the changes in chroma under G hue-high-value conditions (G1 × G3 × G5), the main effect of chroma changes under green hue and high-value conditions was significant for hopeless–hopeful (F = 4.908, *p* < 0.05), annoying–enjoyable (F = 3.372, *p* < 0.05), depressed–cheerful (F = 5.358, *p* < 0.01), and boring–fun (F = 4.972, *p* < 0.05) vocabulary. The main effect by gender and the interaction effect between color and gender was not significant.

As shown in Table 11, in MANOVA (wilks’ λ = 0.542, F(14, 120) = 3.071, η^2^ = 0.264), as per changes in chroma under G hue-low-value conditions (G2 × G4 × G6), the main effect of chroma changes under green hue and low-value conditions was significant for hopeless–hopeful (F = 12.018, *p* < 0.001), annoying–enjoyable (F = 3.374, *p* < 0.05), depressed–cheerful (F = 3.268, *p* < 0.001), and boring–fun (F = 4.560, *p* < 0.05) vocabulary. The main effect according to gender was significant for annoying–enjoyable (F = 4.004, *p* < 0.05) and depressed–cheerful (F = 7.883, *p* < 0.01). The interaction effect between chroma change and gender was insignificant.

As shown in Table 12, in MANOVA (wilks’ λ = 0.524, F(14, 120) = 0.3.266, η^2^ = 0.276), as per changes in chroma under B hue-high-value conditions (B1 × B3 × B5), the main effect of chroma changes under blue hue and high-value conditions was significant for hopeless–hopeful (F = 12.366, *p* < 0.001), annoying–enjoyable (F = 14.748, *p* < 0.001), depressed–cheerful (F = 14.914, *p* < 0.001), boring–fun (F = 9.920, *p* < 0.001), and overwhelming–free (F = 7.084, *p* < 0.01). The main effect according to gender was significant for dislike–like (F = 10.685, *p* < 0.01), and the interaction effect between chroma change and gender was insignificant.

As shown in Table 13, in MANOVA (wilks’ λ = 0.523, F(14, 120) = 3.281, η^2^ = 0.277), as per the changes in chroma under B hue-low-value conditions (B2 × B4 × B6), the main effect of chroma changes under blue hue and high-value conditions was significant in hopeless–hopeful (F = 12.908, *p* < 0.001), annoying–enjoyable (F = 4.694, *p* < 0.05), depressed–cheerful (F = 7.008, *p* < 0.01), boring–fun (F = 16.067, *p* < 0.001), and overwhelming–free (F = 5.947, *p* < 0.01). There was no significant main effect according to gender, and the interaction between chroma change and gender was significant in annoying–enjoyable (F = 3.556, *p* < 0.05).

According to chroma changes, the difference in perceived stress was more intense for blue hues than for green. The main effect of chroma change was more significant than the main effect by gender, indicating a difference in perceived stress in colors according to chroma change. The interaction effect between chroma change and gender was significant for “annoying–enjoyable,” “depressed-cheerful,” and “dislike-like” under green and blue hue and low-value conditions; therefore, we can infer that males and females perceive stress differently because of the influence of chroma in low-value colors.

### 3.2. Discussion

With developments in technology, an increasing number of mobile devices and applications have been used to measure HRV and examine stress levels. Accordingly, parameters for valid HRV measurements taken in under less than 5 min have been developed, such as normalized LF, HF, and RMSSD (Salahuddin et al., 2007 [44], Castaldo et al., 2019 [41], Baek et al., 2015 [45], and Malik et al., 1996 [46]) parameters that were used in this study. According to previous studies, LF reflects sympathetic and parasympathetic nerve activity in short-term measurements, and HF reflects parasympathetic nervous system activity. The main effects of HRV parameters according to changes in chroma and gender were verified. However, the interaction effects between changes in chroma and HRV parameters for males and females were not. This was due to the use of colors in the range of environment colors, which provided a relatively weak stimulus compared to primary colors as the experimental stimulus [23]. Differences between data must be distinct to verify the interaction effect between variables; however, there was no significant change when using colors with weak visual stimulus. Therefore, future research should supplement this limitation. In addition, no interaction effect was observed in short-term HRV measurements; therefore, whether there is an interaction effect between each variable must be investigated through long-term HRV measurements in future studies.

The parameters with statistically significant results were as follows: ln(LF) showed significant results for gender in all changes in environmental color chroma, except for under blue hue-low-value conditions. During chroma changes (G1 × G3 × G5) under green hue-high-value conditions, ln(LF) of males was the highest for G3, but there was no significant change for females. Considering that ln(LF) reflects both sympathetic and parasympathetic activity in short-term measurements and positively correlates with depression, anger, and stress state [47,48,49], it can be determined that stress was stimulated in the case of males for G3. Under green hue-low-value conditions, ln(LF) decreased in both males and females as chroma increased, reflecting a fatigued state. This indicates that although the low-value high-chroma color of the green hue does not stimulate stress, it may give the body a sense of fatigue and reduce physical vitality.

Under blue hue-high-value conditions, ln(LF) of males decreased as chroma increased, whereas ln(LF) of females did not show any change. Blue hue, high-value, and high chroma color were noted to give males a sense of fatigue and slightly reduce their physical vitality. RMSSD of males and females decreased as chroma increased under blue hue-low-value conditions, and the difference was statistically significant. It can be inferred that blue hue, low-value, and high chroma color stimulates stress in males and females.

This study aimed to investigate the difference between physiological stress response through HRV measurement and psychological stress response through color–stress evaluation. The color–stress evaluation results showed statistically significant differences, with the stress emotion perceived through color varying with changes in chroma. The higher the chroma, the more positive the stress emotion was perceived under high-value conditions. The lower the chroma, the more negative the stress emotion was perceived under low-value conditions. These results support the findings of Öztürk et al. [31], Suk et al. [32], and Zielinski [33].

From the color–stress evaluation results, the main effect by gender was observed under G hue-low-value and B hue-low-value conditions. This indicates that under low-value conditions, the stress emotion of males and females according to changes in chroma can be perceived differently. In the HRV measurement results, it was previously noted that both males and females became fatigued under G hue-low-value conditions and were in a stressed state under B hue conditions. Thus, there is a slight difference between the physiological and psychological stress responses of males and females due to changes in chroma under low-value conditions. This demonstrates that when applying low-value colors to a spatial environment, colors should be applied carefully, considering the differences in the stress responses depending on gender.

This research measured HRV due to color–stress in the space where lighting is installed, and depending on the lighting environment, the results may vary. According to Davis et al. [68], while CRI is the only international standardized method that evaluates the coloring ability of a light source when applied to an LED, it has been indicated that the appropriate evaluation regarding coloring rendering could be difficult. Moreover, according to Durmus et al. [69], CCT can lack accuracy in delivering color information. Royer et al. [70] examined an experiment case to investigate the influence of color expression on the perception of an architectural environment. As a result, standardized color reproduction of ANSI/IES TM-30-20 and/or CIE 224:2017 and CIE S026: 2018 should be reported, and the range of color rendering characteristics should be sampled. In addition, it was stated that a minimum population of 30 for the experiment was recommended, but that power analysis was necessary to decide an appropriate sample size. Therefore, the condition of lighting in the color experimental condition is critical, and sufficient consideration is required. Furthermore, it is necessary for statistical analysis to proceed in determining the population needs.

## 4. Conclusions

This study investigated how changes in environmental color chroma affected the HRV and stress emotions of males and females to derive evidence-based research results for recommending colors that have positive effects on the stress levels of occupants and for limiting the use of colors with negative effect on environmental color scheme planning. The main findings were as follows.

First, the interaction effect between changes in environmental color chroma and the HRV of males and females was not verified. This study did not use colors with high visual stimulus as the experimental stimulus; therefore, the differences in the data for each variable and the differences obtained through short-term HRV measurement were insignificant. Future studies should address these matters to validate the interaction effect.

Second, the main effect of HRV parameters for males and females while changing the chroma of environmental colors was verified, and the valid parameters were ln(LF) and RMSSD. Physical vitality decreased for both males and females as they became fatigued due to decreasing ln(LF), as chroma increased under green hue-low-value conditions. Additionally, the RMSSD of males and females decreased as chroma increased under blue hue-low-value conditions. Considering that RMSSD is an indicator that reflects the activity of the parasympathetic nervous system, it can be viewed that the high chroma under blue hue-low-value conditions stimulated stress in males and females. These results show that the use of colors under low-value-high-chroma conditions is highly likely to affect physiological stress; hence, colors with these conditions should be used with care in environmental color schemes.

Third, according to the stress assessment results of environmental colors, the main effect according to chroma change was verified; the higher the chroma in the value range, the more positive the emotions, and the lower the chroma in the low-value range, the more negative the emotions. These results support the findings of previous studies [31,32,33]. However, the stress emotions according to changes in chroma under low-value conditions were perceived differently according to gender. This implies that there may be slight differences between the physiological stress responses by HRV and the psychological stress responses through color evaluation by gender. This finding suggests that low-value colors should be applied with caution in color schemes, as there may be differences in the stress response depending on gender.

This study used and verified HRV measurement research methodology by applying it to the field of environmental color and has significance in proposing a convergent research methodology. However, this methodology is still in the basic research stage, and it is necessary to continuously validate whether HRV parameters respond effectively to color stimuli through further experiments in the future. This scientific research method has the potential to be established as an evidence-based design methodology in the field of environmental color.

## 5. Limitations

In this study, we aimed to assess the stress levels of males and females according to chroma changes in diverse color environments using HRV measurement. While this study has practical significance in that its derived evidence-based research outcomes applicable to the environmental color field, there were some limitations. First, the experimental environment of this study was nearly identical to that of general office spaces. However, as this is a color-related study, a more specialized contemplation of color temperatures and color rendering, which are factors associated with the lighting of the experimental environment, is required. Second, the measuring time of the HRV measurement experiment was minimized considering the psychological stress of the subjects. During this process, HRV was measured for 2 min and 30 s, and the subject performed the experiment in one day. However, it has been suggested that the ideal HRV measuring time is 5 min and that HRV data should be measured repetitively across a designated period to secure more accurate experimental data. In addition, it will be necessary to secure more than 30 people as a population to improve the reliability of research data.

## Figures and Tables

**Figure 1 ijerph-19-05711-f001:**
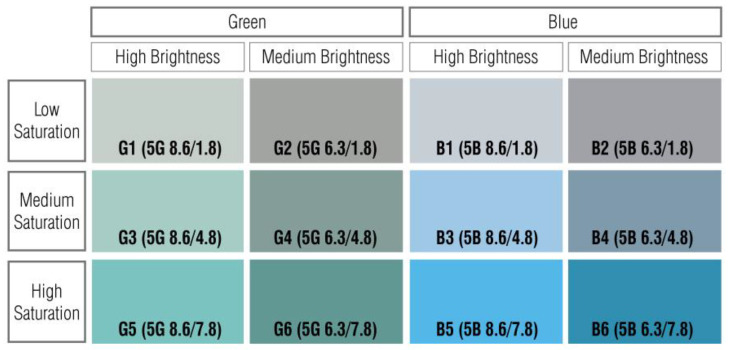
12 Color samples used for the experiment on the HRV.

**Figure 2 ijerph-19-05711-f002:**
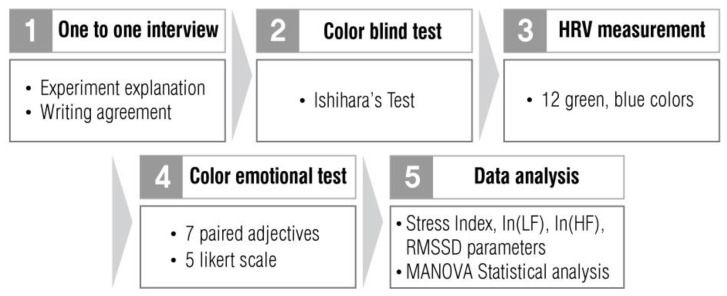
Study design and research procedure.

**Figure 3 ijerph-19-05711-f003:**
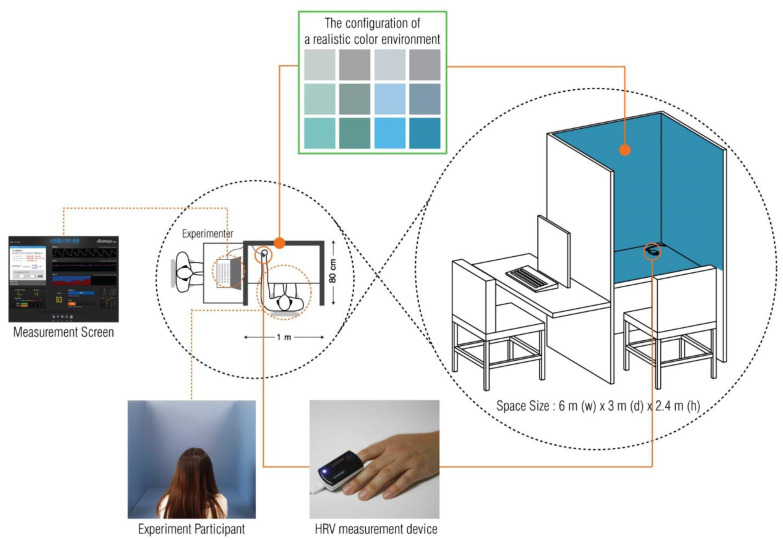
HRV measurement environment.

**Table 1 ijerph-19-05711-t001:** Specifications of the spectrophotometer (Minolta CM-2500d).

Model	Minolta CM-2500d	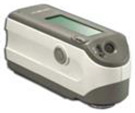
Illumination, viewing system	di:8, de:8 (diffuse illumination, 8-degree viewing), equipped with simultaneous measurement of specular component included (SCI)/specular component excluded (SCE), conforms to CIE No.15, ISO 7724/1, ASTM E1164, DIN 5033 Teil 7, and JIS Z8722 condition C standard	
Wavelength range	360–740 nm	
Wavelength pitch	10 nm	
Reflectance range	0 to 175%, resolution: 0.01%	
Light source	Two pulsed xenon lamps	

**Table 2 ijerph-19-05711-t002:** Data analysis of environmental colors in public health facilities.

Category	Range of Wall Color Data (Min–Max)		Average Color Data	
	Range of Value	Range of Chroma	Value	Chroma
Dominant color	5.3–9.5	0.3–4.2	8.6	1.8
Complementary color	5.2–10.2	0.1–6.6	7.4	2.5
Accent color	3.2–8.5	0.2–10.3	6.3	3.4

**Table 3 ijerph-19-05711-t003:** Measurement device specifications and software application for HRV.

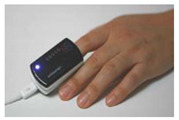		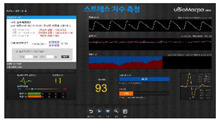	
Model	uBio Macpa	Model	uBio Macpa Software Application
Power	220 V, 60 Hz for computer system and 5V-DC through USB for measurement probe	CPU	Pentium IV 1 GHz above
Power Consumption	0.25 W	OS	Windows 2000 Windows XP, Windows Vista, Windows 7/8/10/11
Dimensions	58 mm (W) × 27 mm (H) × 32 mm (D)	Memory	1 GB above
Measurement Range	40–200 BPM	Disk	100 GB above

**Table 4 ijerph-19-05711-t004:** Stress assessment vocabulary selection process.

	Emotional Vocabulary	Researcher
Doctoral dissertation	Annoyed-enjoyable/noisy-quiet/hopeful-worried/bored-comfortable/looks strong-looks gentle/confident-unconfident/calm-passionate/relaxing-irritating/like-dislike/feel weak-feel strong/oppress-encourage/overwhelmed-free/sleepy-sober/adorable-frustrated/noisy-quiet/depressed–cheerful/firm-soft/easy-nervous	Yu (2009) [54]
	Vivid-subtle/rough-soft/strong-weak/warm-cold/bright-dark/natural-unnatural/clear-dull/not dazzling-dazzling/clear-blurry/light-heavy/pleasant-unpleasant	Seo (2015) [55]
	Happy-unhappy/enjoyable-annoying/hopeful-hopeless/fun-boring/stable-stimulating/calm-excited/relaxed-tensioned/active-static/pleasant-unpleasant/bright-dark/warm-cold/light-heavy/clear-sleepy/confident-afraid/affecting-affected/strong-weak	Ryu (2016) [56]
	Warm-cold/bright-dark/light-heavy/vivid-murky/soft-hard/active-static/like-dislike	Ha (2018) [57]
▼ (Evaluation by three experts) ▼		
Final vocabulary	Hopeless–hopeful/annoying–enjoyable/tensioned–relaxed/depressed–cheerful/boring–fun/overwhelming–free/dislike–like	

**Table 5 ijerph-19-05711-t005:** Specifications of experimental environment artificial lighting (Osram FHF32SS EX-D).

**Model**	**Color Temperature**	**CRI**	**Light Output**	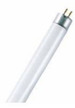
Osram FHF32SS EX-D	6500 K	60	2850 lm	

**Table 6 ijerph-19-05711-t006:** Specifications of photometer (Luton LM-81LX).

**Model**	**Measurement**	**Measurement Range (Accuracy)**		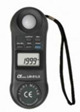
Lutron LM-81LX	Illuminance	Lux	Foot-candle	
		0–20,000 lux (±5% rdg ±8 dgt)	0–2000 Fc (±5% rdg ±8 dgt)	

**Table 7 ijerph-19-05711-t007:** Average ln(LF), ln(HF), and RMSSD parameters for each environmental color.

Green				Blue			
Classification		Male Mean (SD)	Female Mean (SD)	Classification		Male Mean (SD)	Female Mean (SD)
G1 (5G 8.6/1.8)	ln(LF)	7.81 (0.95)	7.55 (0.84)	B1 (5B 8.6/1.8)	ln(LF)	8.34 (0.86)	7.49 (0.64)
	ln(HF)	6.77 (0.75)	6.83 (0.94)		ln(HF)	7.08 (0.68)	6.80 (0.88)
	RMSSD	29.87 (12.18)	35.87 (21.10)		RMSSD	31.26 (12.48)	38.19 (21.13)
G2 (5G 6.3/1.8)	ln(LF)	8.18 (0.84)	7.54 (0.71)	B2 (5B 6.3/1.8)	ln(LF)	8.22 (0.81)	7.59 (0.82)
	ln(HF)	6.79 (0.61)	6.95 (0.72)		ln(HF)	6.71 (0.77)	7.07 (0.81)
	RMSSD	29.44 (14.48)	35.49 (17.05)		RMSSD	29.39 (11.94)	36.95 (18.78)
G3 (5G 8.6/4.8)	ln(LF)	7.87 (0.65)	7.37 (0.74)	B3 (5B 8.6/4.8)	ln(LF)	7.80 (0.94)	7.44 (1.01)
	ln(HF)	6.78 (0.64)	6.78 (0.95)		ln(HF)	6.90 (0.76)	6.77 (1.03)
	RMSSD	29.14 (14.61)	36.07 (24.12)		RMSSD	29.83 (13.31)	34.73 (20.66)
G4 (5G 6.3/4.8)	ln(LF)	8.01 (0.77)	7.68 (0.75)	B4 (5B 6.3/4.8)	ln(LF)	7.83 (1.01)	7.58 (0.90)
	ln(HF)	6.83 (0.63)	6.84 (0.88)		ln(HF)	6.78 (1.02)	6.91 (0.96)
	RMSSD	29.14 (12.24)	35.88 (18.64)		RMSSD	29.16 (13.19)	38.65 (20.99)
G5 (5G 8.6/7.8)	ln(LF)	8.17 (1.15)	7.39 (0.94)	B5 (5B 8.6/7.8)	ln(LF)	7.72 (0.99)	7.45 (0.78)
	ln(HF)	6.81 (0.94)	6.81 (0.86)		ln(HF)	6.68 (0.63)	6.91 (0.85)
	RMSSD	30.39 (14.34)	36.40 (19.32)		RMSSD	30.14 (13.66)	37.49 (20.21)
G6 (5G 6.3/7.8)	ln(LF)	7.62 (0.63)	7.18 (1.09)	B6 (5B 6.3/7.8)	ln(LF)	7.64 (0.82)	7.57 (0.90)
	ln(HF)	6.64 (0.66)	6.63 (1.05)		ln(HF)	6.77 (0.83)	6.73 (0.97)
	RMSSD	28.39 (12.92)	33.80 (21.42)		RMSSD	27.54 (11.69)	34.04 (20.89)

**Table 8 ijerph-19-05711-t008:** Effect and interaction by chroma changes and gender for ln(LF), ln(HF), and RMSSD.

Color	Variable	Sub-Factors	Sum of Squares	Degrees of Freedom	Mean Square	F	*p*
Before × G1 × G3 × G5	Color	ln(LF)	1.759	3	0.586	0.992	0.401
		ln(HF)	0.405	3	0.135	0.220	0.882
		RMSSD	365.440	3	121.813	0.365	0.778
	Gender	ln(LF)	2.761	1	2.761	4.667	0.033 *
		ln(HF)	0.747	1	0.747	1.220	0.272
		RMSSD	1032.256	1	1032.256	3.095	0.082
	Color × Gender	ln(LF)	1.308	3	0.436	0.737	0.533
		ln(HF)	0.870	3	0.290	0.474	0.701
		RMSSD	16.675	3	5.558	0.017	0.997
	Error	ln(LF)	52.050	88	0.591		
		ln(HF)	53.870	88	0.612		
		RMSSD	29,351.826	88	333.543		
Before × G2 × G4 × G6	Color	ln(LF)	3.307	3	1.102	1.459	0.231
		ln(HF)	1.109	3	0.370	0.531	0.662
		RMSSD	474.934	3	158.311	0.515	0.673
	Gender	ln(LF)	3.388	1	3.388	4.484	0.037 *
		ln(HF)	0.352	1	0.352	0.505	0.479
		RMSSD	971.210	1	971.210	3.162	0.079
	Color × Gender	ln(LF)	1.693	3	0.564	0.747	0.527
		ln(HF)	1.106	3	0.369	0.529	0.663
		RMSSD	23.019	3	7.673	0.025	0.995
	Error	ln(LF)	66.501	88	0.756		
		ln(HF)	61.284	88	0.696		
		RMSSD	27,025.290	88	307.106		
Before× B1 × B3 × B5	Color	ln(LF)	3.287	3	1.096	1.643	0.185
		ln(HF)	0.188	3	0.063	0.094	0.963
		RMSSD	324.959	3	108.320	0.349	0.790
	Gender	ln(LF)	4.824	1	4.824	7.233	0.009 **
		ln(HF)	0.298	1	0.298	0.447	0.506
		RMSSD	1064.336	1	1064.336	3.431	0.067
	Color × Gender	ln(LF)	2.252	3	0.751	1.125	0.343
		ln(HF)	2.431	3	0.810	1.216	0.309
		RMSSD	33.331	3	11.110	0.036	0.991
	Error	ln(LF)	58.699	88	0.667		
		ln(HF)	58.642	88	0.666		
		RMSSD	27,300.225	88	310.230		
Before × B2 × B4 × B6	Color	ln(LF)	0.640	3	0.213	0.297	0.827
		ln(HF)	0.492	3	0.164	0.224	0.879
		RMSSD	475.648	3	158.549	0.493	0.688
	Gender	ln(LF)	0.504	1	0.504	0.702	0.404
		ln(HF)	1.008	1	1.008	1.378	0.244
		RMSSD	1391.613	1	1391.613	4.326	0.040 *
	Color × Gender	ln(LF)	0.292	3	0.097	0.135	0.939
		ln(HF)	0.870	3	0.290	0.397	0.756
		RMSSD	27.455	3	9.152	0.028	0.993
	Error	ln(LF)	63.207	88	0.718		
		ln(HF)	64.354	88	0.731		
		RMSSD	28,307.503	88	321.676		

** *p* < 0.01, * *p* < 0.05.

**Table 9 ijerph-19-05711-t009:** Average values of stress assessment for males and females according to changes in environmental color chroma.

**G1 × G3 × G5** **(Green-High-Value)**	**G2 × G4 × G6** **(Green-Low-Value)**
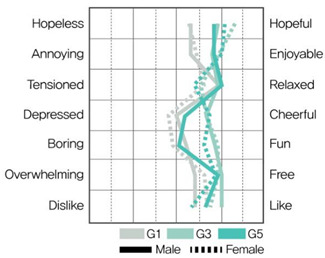	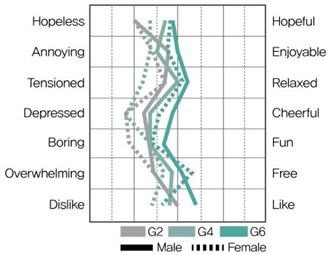
**B1 × B3 × B5** **(Blue-high-value)**	**B2 × B4 × B6** **(Blue-low-value)**
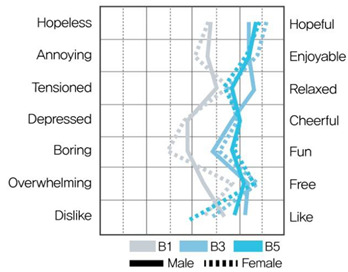	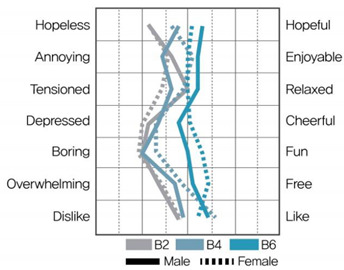

**Table 10 ijerph-19-05711-t010:** Main and interaction effects of stress according to chroma changes under green hue and high-value conditions.

Color	Variable	Sub-Factors	Sum of Squares	Degrees of Freedom	Mean Square	F	*p*
G1 × G3 × G5	Color	Hopeless–hopeful	7.622	2	3.811	4.908	0.010 *
		Annoying–enjoyable	3.846	2	1.923	3.372	0.040 *
		Tensioned–relaxed	0.024	2	0.012	0.016	0.984
		Depressed–cheerful	7.180	2	3.590	5.358	0.007 **
		Boring–fun	6.369	2	3.184	4.972	0.010 *
		Overwhelming–free	2.239	2	1.119	1.472	0.237
		Dislike–like	1.757	2	0.879	1.945	0.151
	Gender	Hopeless–hopeful	0.675	1	0.675	0.869	0.355
		Annoying–enjoyable	0.370	1	0.370	0.649	0.423
		Tensioned –relaxed	2.904	1	2.904	3.934	0.051
		Depressed–cheerful	0.181	1	0.181	0.271	0.605
		Boring–fun	0.156	1	0.156	0.244	0.623
		Overwhelming–free	0.000	1	0.000	0.000	1.000
		Dislike–like	0.268	1	0.268	0.592	0.444
	Color × Gender	Hopeless–hopeful	0.622	2	0.311	0.401	0.671
		Annoying–enjoyable	0.069	2	0.034	0.060	0.942
		Tensioned –relaxed	0.302	2	0.151	0.204	0.816
		Depressed–cheerful	1.513	2	0.756	1.129	0.330
		Boring–fun	1.702	2	0.851	1.329	0.272
		Overwhelming–free	0.239	2	0.119	0.157	0.855
		Dislike–like	1.035	2	0.518	1.145	0.324
	Error	Hopeless–hopeful	51.244	66	0.776		
		Annoying–enjoyable	37.644	66	0.570		
		Tensioned–relaxed	48.711	66	0.738		
		Depressed–cheerful	44.222	66	0.670		
		Boring–fun	42.267	66	0.640		
		Overwhelming–free	50.178	66	0.760		
		Dislike–like	29.822	66	0.452		

** *p* < 0.01, * *p* < 0.05.

**Table 11 ijerph-19-05711-t011:** Main and interaction effects of stress according to chroma changes under green hue and low-value conditions.

Color	Variable	Sub-Factors	Sum of Squares	Degrees of Freedom	Mean Square	F	*p*
G2 × G4 × G6	Color	Hopeless–hopeful	8.724	2	4.362	12.018	0.000 ***
		Annoying–enjoyable	2.022	2	1.011	3.374	0.040 *
		Tensioned –relaxed	3.969	2	1.984	2.039	0.138
		Depressed–cheerful	6.535	2	3.268	8.871	0.000 ***
		Boring–fun	2.467	2	1.233	2.120	0.128
		Overwhelming–free	6.080	2	3.040	4.560	0.014 *
		Dislike–like	2.502	2	1.251	1.841	0.167
	Gender	Hopeless–hopeful	0.237	1	0.237	0.653	0.422
		Annoying–enjoyable	1.200	1	1.200	4.004	0.049 *
		Tensioned–relaxed	1.712	1	1.712	1.759	0.189
		Depressed–cheerful	2.904	1	2.904	7.883	0.007 **
		Boring–fun	1.008	1	1.008	1.733	0.193
		Overwhelming–free	0.004	1	0.004	0.006	0.941
		Dislike–like	2.223	1	2.223	3.272	0.075
	Color × Gender	Hopeless–hopeful	0.391	2	0.195	0.538	0.586
		Annoying–enjoyable	0.022	2	0.011	0.037	0.964
		Tensioned –relaxed	2.024	2	1.012	1.040	0.359
		Depressed–cheerful	0.035	2	0.018	0.048	0.953
		Boring–fun	0.467	2	0.233	0.401	0.671
		Overwhelming–free	0.413	2	0.206	0.310	0.735
		Dislike–like	1.724	2	0.862	1.269	0.288
	Error	Hopeless–hopeful	23.956	66	0.363		
		Annoying–enjoyable	19.778	66	0.300		
		Tensioned –relaxed	64.222	66	0.973		
		Depressed–cheerful	24.311	66	0.368		
		Boring–fun	38.400	66	0.582		
		Overwhelming–free	44.000	66	0.667		
		Dislike–like	44.844	66	0.679		

**** p* < 0.001, ** *p* < 0.01, * *p* < 0.05.

**Table 12 ijerph-19-05711-t012:** Main and interaction effects of stress according to chroma changes under blue hue and low-value conditions.

Color	Variable	Sub-Factors	Sum of Squares	Degrees of Freedom	Mean Square	F	*p*
B1 × B3 × B5	Color	Hopeless–hopeful	14.506	2	7.253	12.366	0.000 ***
		Annoying–enjoyable	17.002	2	8.501	14.748	0.000 ***
		Tensioned–relaxed	1.906	2	0.953	1.066	0.350
		Depressed–cheerful	16.772	2	8.386	14.914	0.000 ***
		Boring–fun	15.591	2	7.795	9.920	0.000 ***
		Overwhelming–free	8.357	2	4.179	7.084	0.002 **
		Dislike–like	1.672	2	0.836	1.047	0.357
	Gender	Hopeless–hopeful	0.408	1	0.408	0.696	0.407
		Annoying–enjoyable	0.023	1	0.023	0.040	0.842
		Tensioned –relaxed	0.533	1	0.533	0.597	0.443
		Depressed–cheerful	0.033	1	0.033	0.059	0.808
		Boring–fun	0.112	1	0.112	0.143	0.707
		Overwhelming–free	1.556	1	1.556	2.639	0.109
		Dislike–like	8.533	1	8.533	10.685	0.002 **
	Color × Gender	Hopeless–hopeful	0.506	2	0.253	0.431	0.652
		Annoying–enjoyable	0.557	2	0.279	0.483	0.619
		Tensioned–relaxed	1.739	2	0.869	0.973	0.383
		Depressed–cheerful	0.106	2	0.053	0.094	0.911
		Boring–fun	0.980	2	0.490	0.623	0.539
		Overwhelming–free	0.635	2	0.318	0.538	0.586
		Dislike–like	0.672	2	0.336	0.421	0.658
	Error	Hopeless–hopeful	38.711	66	0.587		
		Annoying–enjoyable	38.044	66	0.576		
		Tensioned –relaxed	58.978	66	0.894		
		Depressed–cheerful	37.111	66	0.562		
		Boring–fun	51.867	66	0.786		
		Overwhelming–free	38.933	66	0.590		
		Dislike–like	52.711	66	0.799		

**** p* < 0.001, ** *p* < 0.01.

**Table 13 ijerph-19-05711-t013:** Main effect and interaction effect of stress according to chroma changes under blue hue and high-value conditions.

Color	Variable	Sub-Factors	Sum of Squares	Degrees of Freedom	Mean Square	F	*p*
B2 × B4 × B6	Color	Hopeless–hopeful	13.891	2	6.945	12.908	0.000 ***
		Annoying–enjoyable	3.439	2	1.719	4.694	0.012 *
		Tensioned–relaxed	2.069	2	1.034	1.099	0.339
		Depressed–cheerful	8.985	2	4.493	7.008	0.002 **
		Boring–fun	17.917	2	8.958	16.067	0.000 ***
		Overwhelming–free	9.580	2	4.790	5.947	0.004 **
		Dislike–like	3.563	2	1.781	2.601	0.082
	Gender	Hopeless–hopeful	0.237	1	0.237	0.441	0.509
		Annoying–enjoyable	0.300	1	0.300	0.819	0.369
		Tensioned–relaxed	0.579	1	0.579	0.615	0.436
		Depressed–cheerful	0.023	1	0.023	0.036	0.850
		Boring–fun	0.408	1	0.408	0.732	0.395
		Overwhelming–free	0.890	1	0.890	1.105	0.297
		Dislike–like	0.334	1	0.334	0.488	0.487
	Color × Gender	Hopeless–hopeful	0.169	2	0.084	0.157	0.855
		Annoying–enjoyable	2.606	2	1.303	3.556	0.034 *
		Tensioned–relaxed	2.124	2	1.062	1.129	0.330
		Depressed–cheerful	0.541	2	0.270	0.422	0.658
		Boring–fun	0.417	2	0.208	0.374	0.690
		Overwhelming–free	0.246	2	0.123	0.153	0.859
		Dislike–like	2.341	2	1.170	1.709	0.189
	Error	Hopeless–hopeful	35.511	66	0.538		
		Annoying–enjoyable	24.178	66	0.366		
		Tensioned–relaxed	62.089	66	0.941		
		Depressed–cheerful	42.311	66	0.641		
		Boring–fun	36.800	66	0.558		
		Overwhelming–free	53.156	66	0.805		
		Dislike–like	45.200	66	0.685		

**** p* < 0.001, ** *p* < 0.01, * *p* < 0.05.

## Data Availability

Not available.

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
