# Peer review of "Effects of Changes in Environmental Color Chroma on Heart Rate Variability and Stress by Gender"

_ijerph, 2022, doi:10.3390/ijerph19095711_

Round 1

Reviewer 1 Report

Dear Authors, 

The aim of this study was to assess stress level in different color environments using HRV analysis in men and women. The authors continue to study stress levels in colorful environments. The article has a practical meaning. However, it has serious flaws in methodology. The manuscript has low integrity. The purpose of study (lines 134-138) is unclear. Poor English makes it difficult to read and understand the text. The style and grammar need careful editing.

Authors should more clearly formulate the purpose of study. The Methodology section needs to be improved. Subjects characteristics, study groups, tests points, output measurements (methods of stress level assessment and ECG registration, HRV analysis), and plan of statistical analysis should be more clearly presented.

Paragraph 2.6. HRV measurement environment contains the information about configuration of experimental colors (lines 318-335), that could be placed to paragraph 2.1. Lines 336-346 could be placed to paragraph 2.3.

The results of HRV in Table 6 are presented without units of measurement. What do numbers like “7.81(0.95)" mean?

Authors should use unified terms for sex/genders groups: “male-female” or “man-woman”.

The abbreviation of the OECD has not been explained.

The study has complicated design. The results should be presented more clearly for a better understanding of the main results of the study.

My conclusion, the article could not be published as it has serious flaws in methodology, additional experiments needed, research not conducted correctly.

Author Response

Thank you for your comments for the article. Please check to the attached file for the answer. 

Reviewer 2 Report

The study is very interesting from different perspectives. However, in its current state, it requires various adjustments and corrections. I must indicate that the version I review has adjustments indicated, perhaps by other reviewers or the authors, which are indicated in blue (review sent version). However, the authors should review the following points with a focus on statistical analyses:

1) Line 300, the total number of participants is indicated (n=24, 9 males and 15 females), but when reviewing tables 7, 9, 10, 11, and 12 of the MANOVAs in the results they indicate in the column of degrees of freedom specifically in the error of the bifactorial model 88 and 66 df. This is not correct because it should be the sum of the degrees of freedom and a number less than the total number of participants (n=24). So, is your analysis bloated on "replications" or do you have pseudoreplication? They must explain and clarify this serious mistake.
2) Associated with MANOVA, it is not indicated that they verified the normal error distribution assumptions, nor the homogeneity of variances, which are requirements of the statistical test. For example, the Likert scale is of ordinal type the variable, which often suggests that it will not have a normal error distribution. In addition, the MANOVA must be associated with reporting the Wilks Lambdas, values, ​​and associated probabilities that are not reported in the results. In addition, there are parts in the text where they indicate p<0.0000 (line 435 and 436, 444, 445, table 10, for example) which is an error since it must be limited to a value of p<0.001, p<0.00001, for example. I suggest consulting a statistician.

Therefore, these two points that represent the argument of the results are very weak and in the current form that the manuscript is in, it requires a major revision in the analysis and results. Obviously, if they make the appropriate adjustments, the response pattern may not vary, but committing a type II statistical error would place them in the possibility of accepting false hypotheses as true with the results reported in the current version of the manuscript.

Author Response

(The authors gave the same response as above.)

Reviewer 3 Report

The study aims to analyze the effect of color samples on HRV. In that sense, it is a niche study and potentially a useful contribution to the literature. There are some weaknesses that should be strengthened to improve the quality of the manuscript.

Color of surfaces is a combination of light source spectra and surface reflectance characteristics. The authors can report the lighting a little better (e.g., report Duv, IES-TM-30 Rf fidelity and Rg gamut indices, show the spectral power distribution of the lamp in a figure). To complete the analysis, authors should discuss the importance of correlated color temperature (CCT), Duv, and color rendering metrics, such as ANSI/IES TM-30, CRI Ra, and acknowledge their well-documented limitations. Authors can refer to

  • Davis, W., & Ohno, Y. (2009). Approaches to color rendering measurement. Journal of Modern Optics, 56(13), 1412-1419.
  • Durmus, D. (2021). Correlated color temperature: Use and limitations. Lighting Research & Technology, 14771535211034330.
  • ANSI/IES TM-30-20 IES Method for Evaluating Light Source Color Rendition https://store.ies.org/product/tm-30-20-ies-method-for-evaluating-light-source-color-rendition/

For the accuracy of the HRC and color related studies, I suggest authors to refer to sources below that can strengthen their experimental protocols (including reporting time intervals, transitions, chromatic adaptation, participant info):

  • Schäfer, A., & Vagedes, J. (2013). How accurate is pulse rate variability as an estimate of heart rate variability?: A review on studies comparing photoplethysmographic technology with an electrocardiogram. International journal of cardiology, 166(1), 15-29.
  • Royer, M., Houser, K., Durmus, D., Esposito, T., & Wei, M. (2021). Recommended methods for conducting human factors experiments on the subjective evaluation of colour rendition. Lighting Research & Technology, 14771535211019864.

For the statistical analysis, I recommend the authors to check the normality and equality of variance of data before applying parametric statistical tests, such as MANOVA. Authors should also report effect size for statistically significant conditions.

I also recommend the authors to add a “limitations” paragraph/section as no experiment is perfect or cannot capture every angle of a phenomenon. Acknowledging limitations would make the paper stronger and more transparent.

Minor comments

  • Table 5. Change “lux, foot-candle” to illuminance. Illuminance is the measured quantity; lux is the unit.
  • Figure captions could be a little more self-explanatory. As they stand now, they are too brief.
  • Variables and constants (e.g., p in the p-value) should be italicized.
  • There are some minor grammar issues and typos. I recommend the authors to carefully read their paper before the next round.
  • HRV is defined (spelled out) later in page 3 after its first introduced in page 2.
  • There are some in-text citation formatting mismatches in the new text.

Author Response

(The authors gave the same response as above.)

Round 2

Reviewer 1 Report

Dear Authors, 

The objective of this study was to evaluate the stress levels of males and females in accordance with the various color environments by utilizing HRV analysis. The authors continue to study stress levels in colorful environments. The article has a practical meaning. The manuscript has been improved. However, it still has serious flaws in methodology. The results obtained are of low significance.

The study was performed using old equipment or many years ago. The results of the analysis of HRV are doubtful, since OS Windows 2000 Windows XP, Windows Vista, Windows 7/8 have not been supported for a long time. Statement of the problem artificially uses the current situation with COVID-19.

Since no valuable results were obtained on HRV parameters in various color environments, the title of the article should be changed, as it does not correspond to the main results. The interaction between stress level and some color environment characteristics was not confirmed by HRV results.

My major concern is that the Authors did not get valuable results on the HRV parameters in different color environments (table 8). This may be due to two main reasons: 1) short ECG recordings (2 minutes 30 seconds) were used to assess HRV, 2) an extremely short break between color tests (1 minute), which could preserve a trace of autonomic reactions to a stimulus. The desire to reduce the duration of the general experimental session, which causes fatigue of the subjects, should not be carried out to the detriment of the quality of registration of individual indicators.

There are many errors in the list of references. Incorrect reference [47]:

Heart rate variability: standards of measurement, physiological interpretation and clinical use. Task Force of the European Society of Cardiology and the North American Society of Pacing and Electrophysiology. Circulation. 1996;93(5):1043-1065.

My conclusion, the article could not be published as it has serious flaws in methodology, additional experiments needed, research not conducted correctly.

Author Response

Thank you for the review comments. They have significantly helped improve the quality of the article. Please checked the attached file. 

Reviewer 2 Report

Dear authors,
The adjustments and corrections that you applied to the manuscript allow more clarity in your statistical procedures and I congratulate you for your attention. However, the only point that I do not notice in the new version is that they do not indicate that they have verified the normal error distribution of the response variables. I understand that the homogeneity of variances justifies it and the way in which they indicate it is adequate, but in the same sense, it should be indicated for the normal error distribution. It is very important since ordinal type response variables are involved.

Author Response

(The authors gave the same response as above.)

Reviewer 3 Report

Color of surfaces is a combination of light source spectra and surface reflectance characteristics. The authors can report the lighting a little better (e.g., report Duv, IES-TM-30 Rf fidelity and Rg gamut indices, show the spectral power distribution of the lamp in a figure). To complete the analysis, authors should discuss the importance of correlated color temperature (CCT), Duv, and color rendering metrics, such as ANSI/IES TM-30, CRI Ra, and acknowledge their well-documented limitations. Authors can cite:

  • Davis, W., & Ohno, Y. (2009). Approaches to color rendering measurement. Journal of Modern Optics, 56(13), 1412-1419.
  • Durmus, D. (2021). Correlated color temperature: Use and limitations. Lighting Research & Technology, 14771535211034330.
  • ANSI/IES TM-30-20 IES Method for Evaluating Light Source Color Rendition https://store.ies.org/product/tm-30-20-ies-method-for-evaluating-light-source-color-rendition/

For the accuracy of the HRC and color related studies, I suggest authors to refer to sources below that can strengthen their experimental protocols (including reporting time intervals, transitions, chromatic adaptation, participant info):

  • Schäfer, A., & Vagedes, J. (2013). How accurate is pulse rate variability as an estimate of heart rate variability?: A review on studies comparing photoplethysmographic technology with an electrocardiogram. International journal of cardiology, 166(1), 15-29.
  • Royer, M., Houser, K., Durmus, D., Esposito, T., & Wei, M. (2021). Recommended methods for conducting human factors experiments on the subjective evaluation of colour rendition. Lighting Research & Technology, 14771535211019864.

Author Response

Thank you for the review comments. They have significantly helped improve the quality of the article. Please checked the attached file. 

This manuscript is a resubmission of an earlier submission. The following is a list of the peer review reports and author responses from that submission.

Round 1

Reviewer 1 Report

General Comments

This study deals with a subject that has already been approached by other investigators, effects of environmental color on stress. The novel addition this study makes to the field is the investigation of effects of color saturation levels on heart rate variability. Overall the manuscript is difficult to follow, the results and conclusions are convoluted, and there is no particular take-home massage that would be useful to readers or other investigators.  There are gaping holes in the methodology regarding the measurement of HRV. In addition, the interpretation of the results is flawed at a very basic level.

Specific Comments 

Abstract. l 18-20. "Consequently the interaction... change was verified." Not clear. Should start with main effect.

l25. "Stress levels changed positively." Does this mean they got more intense or was stress reduced?

Intro. l61. "Stress in the alpha wave.." Explain

l62 "red stimulates ANS" Do you mean the sympathetic branch of ANS? This is an occurring question.

Methods, section 2.1, l159. Define the difference between brightness and saturation.

HRV measurement, l189. On what basis was the exposure time of 2.5 min and resting time of 1 min chosen?

Was the HRV monitor on for the duration of the experiment? How long was the portion of HRV recording that was analysed for each color set-up, 2.5 min? 

Details of the HRV monitor are totally lacking. Fig. 4 shows an HRV monitor on a finger. How accurate is this device? Has it been tested against a chest strap HRV sensor? 

l204. Explain the physiological interpretation of RMSSD, lnLF, lnHF

Table 3. Cannot see the selected combinations "anxious-relaxed" and "overwhelm-free" in the examples from the dissertations.

Figure 4 is unnecessary.

l238. State ethnicity of participants.

Results. Table 6 labels all results as high brightness (8.6). The low brightness labels (6.3) are missing. This is an egregious error.

Table 8 is totally incomprehensible. There is no key regarding the green versus gray plots. Cannot tell which plot represents which color saturation.

l385. The LF range of HRV includes contributions from both the sympathetic and parasympathetic branches of ANS, not just the sympathetic branch.

l424. "Increases in lnLF and RMSSD parameters are viewed as promoting physical vitality by increasing arousal." No - that might be true for lnLF but not for RMSSD. RMSSD reflects the parasympathetic contribution to HRV, which is associated with relaxation, not arousal.

l450. "Decreases in lnLF and RMSSD are positive responses to stress due to a more relaxed physical state." NO- decreases in RMSSD indicate a less relaxed state.

Reviewer 2 Report

The aim of this study was to assess HRV parameters in different color environments in men and women. The authors continue to study stress levels in colorful environments. The article has a practical meaning. However, it has flaws in methodology. The structure of the Manuscript is not appropriate. For the future, authors should improve the text and keep to the structure. All parts of the manuscript contain paragraphs which stay far from the scope of study.

In the "Introduction" section, authors should focus on the purpose of the research and the research methodology should be clearly presented. No information of gender stress reactivity differences was presented in the Introduction section. So, the choice of gender specific study groups was not explained. The use of green and blue environmental colors seems causeless.

Analysis of heart rate variability was performed without using conventional approaches (lines 192-199). Thus, it can be viewed with significant limitations. In lines 97-106, the physiological basement of HRV and references on its application in similar studies should be clearly presented.

Paragraph 2.1. Please, concentrate on the methodology of configuration of experimental colors. The color environment significance (lines 113-130) should be described in the Introduction section.

Lines 142-158: it is unclear, what was the sense of presenting color data collection in respect with configuration of experimental colors.

Paragraph 2.4. HRV measurement experiment – No HRV measurements were presented.

The plan of statistical analysis was not presented.

The Results section should be improved. The study has complicated design. It is difficult to catch main results. It seems that gender differences were found unexpectedly, besides the statistical analysis plan. The interpretation of the HRV results is weak. There are mistakes in the assessment of RMSSD changes. In fact, RMSSD is a marker of parasympathetic activity.

My conclusion, the article could not be published as it has serious flaws, additional experiments needed, research not conducted correctly.

Reviewer 3 Report

The article is a very interesting and valuable scientific work. It has been perfectly scientifically developed. The whole argument is presented factually and logically. The authors' decisions are very precisely explained, clearly showing the scientific strategy and aims. Likewise, the research results are very clear and lead to a well-written conclusion.

My congratulations on a very good article.